# Rapamycin Treatment Alleviates Chronic GVHD-Induced Lupus Nephritis in Mice by Recovering IL-2 Production and Regulatory T Cells While Inhibiting Effector T Cells Activation

**DOI:** 10.3390/biomedicines11030949

**Published:** 2023-03-20

**Authors:** Jilu Zhang, Xun Wang, Renxi Wang, Guojiang Chen, Jing Wang, Jiannan Feng, Yan Li, Zuyin Yu, He Xiao

**Affiliations:** 1State Key Laboratory of Toxicology and Medical Countermeasures, Beijing Institute of Pharmacology and Toxicology, Beijing 100850, China; 2Department of Biomedicine, Institute of Frontier Medical Sciences, Jilin University, Changchun 130021, China; 3Department of Experimental Hematology and Biochemistry, Beijing Key Laboratory for Radiobiology, Beijing Institute of Radiation Medicine, Beijing 100850, China; 4Laboratory of Brain Disorders, Collaborative Innovation Center for Brain Disorders, Beijing Institute of Brain Disorders, Capital Medical University, Ministry of Science and Technology, Beijing 100054, China

**Keywords:** cGVHD, lupus nephritis, rapamycin, Tregs, IL-2

## Abstract

In this study, we test the therapeutic effects of rapamycin in a murine model of SLE-like experimental lupus nephritis induced by chronic graft-versus-host disease (cGVHD). Our results suggest that rapamycin treatment reduced autoantibody production, inhibited T lymphocyte and subsequent B cell activation, and reduced inflammatory cytokine and chemokine production, thereby protecting renal function and alleviating histological lupus nephritis by reducing the occurrence of albuminuria. To explore the potential mechanism of rapamycin’s reduction of kidney damage in mice with lupus nephritis, a series of functional assays were conducted. As expected, rapamycin remarkably inhibited the lymphocytes’ proliferation within the morbid mice. Interestingly, significantly increased proportions of peripheral CD4+FOXP3+ and CD4+CD25high T cells were observed in rapamycin-treated group animals, suggesting an up-regulation of regulatory T cells (Tregs) in the periphery by rapamycin treatment. Furthermore, consistent with the results regarding changes in mRNA abundance in kidney by real-time PCR analysis, intracellular cytokine staining demonstrated that rapamycin treatment remarkably diminished the secretion of Th1 and Th2 cytokines, including IFN-γ, IL-4 and IL-10, in splenocytes of the morbid mice. However, the production of IL-2 from splenocytes in rapamycin-treated mice was significantly higher than in the cells from control group animals. These findings suggest that rapamycin treatment might alleviate systemic lupus erythematosus (SLE)-like experimental lupus nephritis through the recovery of IL-2 production, which promotes the expansion of regulatory T cells while inhibiting effector T cell activation. Our studies demonstrated that, unlike other commonly used immunosuppressants, rapamycin does not appear to interfere with tolerance induction but permits the expansion and suppressive function of Tregs in vivo.

## 1. Introduction

Chronic graft-versus-host disease (cGVHD) is the major late complication following allogeneic transplantation, affecting up to 80% of patients in some series [1]. Compared with acute GVHD, which is more likely to reflect cell apoptosis and necrosis, cGVHD is largely an inflammatory and fibrotic process [1,2,3]. Clinically, cGVHD is characterized by sustained B cell activation and autoantibody production that may lead to renal pathology, chronic liver involvement, and symptoms similar to those of autoimmune disease in allogeneic stem cell transplantation [4,5,6]. Injection of DBA/2 mouse lymphocytes to C57BL/6-DBA/2 F1 hybrids leads to cGVHD characterized by a persistent lymphoid hyperplasia producing hypergammaglobulinemia and a systemic lupus erythematosus (SLE)-like disease [1,7,8] with splenomegaly, B cell expansion, autoantibodies, and severe immune-complex-mediated glomerulonephritis that results in death from lupus-nephritis-induced renal failure [6,9,10,11].

The immunophilin ligand rapamycin, a macrocyclic triene antibiotic produced by *Streptomyces hygroscopicus*, exhibits potent immunosuppressive properties and is used therapeutically to prevent allograft rejection [12,13,14,15,16,17]. It acts by binding to the mammalian target of rapamycin (mTOR)–FK506 binding protein complex, resulting in the inhibition of mTOR protein kinase activity. It is known that rapamycin affects a variety of cell functions, including cell-cycle control, cell growth, apoptosis, transcription and translation efficiency [18,19,20,21]. In this study, we evaluated the therapeutic effect of rapamycin in a murine model of SLE-like experimental lupus nephritis induced by cGVHD and investigated the potential mechanism by which rapamycin regulates the progression of lupus nephritis.

## 2. Materials and Methods

### 2.1. Induction of Lupus Nephritis in cGVHD Model

Female (C57BL/6-DBA/2J) F1 hybrid mice aged 7 to 8 weeks were used as recipients of female DBA/2J donor lymphocytes. The cell suspension containing the mixture of donor cells from thymus, spleen and lymph nodes was injected intravenously (i.v.) four times within two weeks at intervals of 3 to 4 days. Serum and urine samples were collected weekly. At 12–15 weeks after treatment with rapamycin, all surviving animals were sacrificed for tissue harvesting. All animals were purchased from the Beijing Laboratory Animal Center and kept in specific-pathogen-free facilities.

The care, use and treatment of mice in this study were in strict agreement with the guidelines for the care and use of laboratory animals established by the Institute of Basic Medical Sciences.

### 2.2. Rapamycin Treatment

Mice that received DBA/2J donor lymphocytes were randomized to receive either rapamycin (3 mg/kg, *n* = 13) or equal volume of saline as control treatment group (*n* = 13) once daily by oral gavage post transplantation, which were successive until the end of the experiments. Oral liquid formulation of rapamycin was purchased from NCPC New Drug Research and Development Co., Ltd. (Amman, Jordan).

### 2.3. Proteinuria

The urine was collected from spontaneous urination when handling the mice. If mice did not urinate, gentle pressure was applied on the mouse bladder. The urine was collected in 1.5 mL Eppendorf tubes and kept on ice until testing [22,23]. Urinary protein concentration was determined by colorimetric analysis using Bradford protein assay kit (ThermoFisher Scientific, Waltham, MA, USA) within 2 h of collection. The onset of proteinuria was determined by stable urine protein concentration of 100 mg/dL or higher [22]. When the mice showed onset of proteinuria, an additional two proteinuria tests were performed in the following 24 h to avoid any false-positive results.

### 2.4. ELISA

The mouse blood was collected by cutting the tail. The serum was collected by centrifuging the blood at 1000–2000× *g* for 10 min in a refrigerated centrifuge. The standard ELISA assay was used to measure the production of serum autoantibodies 3 weeks after cell injection.

Briefly, 96-well flat-bottom ELISA plates (CorningCostar, Glendale, AZ, USA) were coated with double-stranded DNA (dsDNA) (Sigma-Aldrich, St. Louis, MO, USA) or single-stranded DNA (ssDNA) (Sigma-Aldrich, St. Louis, MO) at 4 °C overnight and then blocked with bovine serum albumin (BSA) (Sigma-Aldrich, St. Louis, MO, USA) before incubation with various dilutions (1:10–1:25) of serum samples from experimental animals. Plates were washed and then incubated with HRP-coupled goat anti-mouse IgG or murine IgG isotype including IgG1 and IgG2a. After incubation with the substrate TMB, the plates were read at O.D. 450 nm.

### 2.5. Histology

Kidney specimens cut from normal mice without cell metastasis and experimental mice were fixed with 10% formaldehyde and then embedded in paraffin. Four-micrometer slices were obtained and stained with hematoxylin and eosin.

### 2.6. qPCR

Quantitative SYBRGreen RT-PCR kit (QIAGEN, Germantown, MD, USA) was used for quantitative PCR of the whole kidney tissue. Each sample was amplified in triplicate, and the relative expression data were determined by normalizing the GAPDH expression measured at the same time in the same sample to calculate the fold change in value using the 2^−ΔΔCT^ method (ABI, step-one, Applied Biosystems, Waltham, MA, USA). The expression level of renal mRNA in normal mice was defined as 1. The primers used are shown in Table 1. Dissociation curve analysis was performed to confirm the specific amplification of each primer for each product.

### 2.7. Immunoblotting

Frozen kidney tissue proteins were prepared by homogenization in M2 lysis buffer, which contained 0.5% NP-40, 250 mM NaCl, 20 mM Tris-HCl (pH 7.6), 3 mM ethylene glycol-bis(β-aminoethyl ether)-*N,N,N′,N′*-tetraacetic acid (EGTA) and 3 mM ethylenediaminetetraacetic acid (EDTA). Before use, 10 µg/mL aprotinin, 1 mM dithiothreitol (DTT), 10 mM p-nitrophenyl phosphate, disodium salt (PNPP) and 1 mM Na_3_VO_4_ were freshly added and quantified by Bradford protein assay. The same amounts of samples were separated by 12% sodium dodecyl sulfate–polyacrylamide gel electrophoresis (SDS-PAGE) and transferred to nitrocellulose membrane. The membrane was incubated with 5% skimmed milk in a heat-sealed bag at room temperature for 1 h, and then incubated with the primary antibody of mouse c-Jun N-terminal kinase (JNK), p-JNK, extracellular-signal-regulated kinases (ERKs), p-ERK, p38 mitogen-activated protein kinases (P38), p-P38 (Cell Signaling, Danvers, MA, USA) or glyceraldehyde 3-phosphate dehydrogenase (GAPDH) (CaliBio, Shenzhen, China) at 4 °C overnight, and gently shaken. Then, we added the second antibody conjugated with relative horseradish peroxidase (HRP), and used the chemiluminescence detection kit (Amersham International, Amersham, UK) to generate the signal.

### 2.8. Flow Cytometry Analysis

Single-cell suspensions of peripheral blood leukocytes or spleen cells were prepared 6–8 weeks after cell transfer. All the flow antibodies and staining kits were purchased from eBiosciences, Inc., San Diego, CA. Cell populations were characterized with the following flow antibodies: CD4 (GK1.5), CD8 (53–6.7), CD69 (H1.2F3), CD44 (IM7), CD62L (MEL-14), TCRβ (H57-597) and ICOS (7E.17G9). Intracellular analysis of FoxP3 (FJK-16s) was performed after fixation and permeabilization, using Fix and Perm reagents. For intracellular cytokine staining, peripheral blood T cells or splenocytes were stimulated in vitro at 37 °C with phorbol myristate acetate (PMA) (50 ng/mL; Sigma) and ionomycin (1 μg/mL; Sigma) in the presence of 1 μg/mL brefeldin A. After 4–5 h culture, cells were collected and stained for surface marker (APC-CD4). Then, cells were washed, fixed and permeabilized according to the manufacturer’s datasheet. PE-conjugated antibodies of cytokines including IFN-γ, IL-2, IL-4 and IL-10 (eBioscience) were then added into the cells in permeabilization buffer for intracellular staining. Last, cells were washed in permeabilization buffer and resuspended in staining buffer for analysis on a flow cytometer. At least 15,000 events were collected and analyzed on a FACSCalibur using CellQuest software (BD Biosciences, San Diego, CA, USA). Isotype matching antibodies were used to define marker gating.

### 2.9. Cell Proliferation by CFSE Analysis

The cells (2 × 10^6^) were labeled with 0.5 μM carboxyfluorescein diacetate succinimidyl ester (CFDA-SE, Molecular Probes, Eugene, OR, USA) and activated with 2 mg/mL anti-CD3 mAb (145-2C11, eBiosciences, Inc., San Diego, CA, USA) and 0.5 mg/mL anti-CD28 mAb (BD Pharmingen, San Diego, CA, USA) in complete culture media, RPMI 1640 (ThermoFisher Scientific, Waltham, MA, USA) containing 10% FBS (ThermoFisher Scientific, Waltham, MA, USA). Intracellular esterases cleave the acetate groups, leading to the fluorescent carboxyfluorescein succinimidyl ester (CFSE). Cell division accompanied by CFSE dilution at 72 h after culture was analyzed by flow cytometry. The proportion of CFSE+ cells proliferating in vitro was calculated as below. The number of cells (events) in each cycle (division: *n*) was divided by 2 raised to power n to calculate the percentage of original precursor cells from which they arose. The sum of original precursors from division 1 to 6 represents the number of precursor cells that proliferated. The percent of CFSE+ divided cells was calculated by (no. of precursors that proliferated_1–6_/no. of total precursors_0–6_) × 100. Percentage of suppression in comparison to proliferation of control group lymphocytes was indicated as 100 × (percentage of divided cells of control group percentage of divided cells of rapamycin group)/(percentage of divided cells of control group).

### 2.10. Statistical Analysis

Data were expressed as mean ± standard deviation (SD). Proteinuria was compared by Chi-square test. For other datasets, Shapiro–Wilk test and homogeneity test of variance were first performed. Datasets meet the standard of normal distribution and homogeneity was further analyzed using Student’s *t*-test; otherwise, Mann–Whitney U test was selected. In each case, *p* < 0.05 was defined as statistically significant, and only significant probabilities are shown.

## 3. Results

### 3.1. Rapamycin Treatment Delayed the Onset of Proteinuria, Decreased Serum Autoantibody Levels and Reduced Renal Tissue Damages in Lupus-Nephritis-Bearing Mice

Lupus nephritis is characterized by proteinuria, elevated serum autoantibody levels and renal tissue damages. To investigate the therapeutic effects of rapamycin in lupus nephritis, we used a well-established cGVHD-induced lupus nephritis mouse model by transferring DBA/2J donor lymphocytes in to C57BL/6-DBA/2J F1 hybrid mice. Rapamycin or vehicle control (saline) was administered once daily by oral gavage post transplantation. As shown in Figure 1A, all the control mice had proteinuria at 12 weeks after cell transfer, while only 30.77% of rapamycin-treated mice developed proteinuria. These data suggest that rapamycin treatment significantly delayed the onset of proteinuria. Serum levels of total anti-ssDNA IgG, IgG1 and IgG2a and anti-dsDNA IgG, IgG1 and IgG2a were measured by ELISA 3 weeks after cell injection. As shown in Figure 1B, rapamycin treatment significantly reduced the level of the circulating autoantibodies.

Histological analysis by H&E staining (Figure 2) showed that the kidney tissue of the control group mice showed mesangial hyperplasia, renal tubular dilation with protein cast deposition in the renal tubules, white blood cell infiltration in the perivascular areas at the beginning of proteinuria development and in interstitial areas at the time of disease progression, as well as renal tubular atrophy, crescent formation and glomerular and vascular sclerosis at the end of the disease (Figure 2B). On the contrary, no significant tissue damage was observed in the renal slices of rapamycin-treated animals without albuminuria, including cell infiltration of renal parenchyma and glomerular or interstitial damage (Figure 2C).

All these data suggest that rapamycin treatment ameliorated lupus nephritis in mice.

### 3.2. Rapamycin Treatment Down-Regulated the Expression of Genes Involved in Pathogenesis of Lupus Nephritis and Inflammation-Related Signal Pathways in the Kidney

In order to determine several pathogenic factors involved in the pathogenesis of lupus nephritis in this human-like SLE mouse model, 36 genes, including inflammatory mediators and fibrosis molecular reverse transcripts from renal mRNA, were analyzed by real-time quantitative PCR. Compared with normal animals, 21 genes were up-regulated in nephrotic mice. Then, the effects of rapamycin treatment on the expression of these 21 genes in the kidney were tested. The results shown in Figure 3A demonstrate that rapamycin treatment significantly down-regulated the mRNA expression of these genes, including: 1. chemokines such as B lymphocyte chemoattractant (BLC), regulated on activation, normal T cell expressed and secreted (RANTES), monocyte chemoattractant protein 1 (MCP-1) and interferon gamma-induced protein 10 (IP-10); 2. proinflammatory cytokines such as TNF-α, IL-1β and IL-6; 3. fiber-forming factors such as TGF-β1 and pro (α1) I collagen; 4. cytokines such as IFN-γ, IL-21, IL-4, IL-10 and T cell immunoglobulin domain and mucin domain 3 (Tim-3); and 5. complement-activated components such as C3a, C5a, C3, C5, C3aR and C5aR. Unexpectedly, the level of IL-2 mRNA expression in rapamycin-treated mice was even higher than that in control mice. It is worth noting that we only tested the mRNA levels of these pathogenic factors, and the qPCR results might not accurately correlate with protein levels of these factors. To better understand the mechanism by which rapamycin regulates these pathogenic factors, the protein levels of these factors need to be confirmed in a future study.

Three members of MAPK, extracellular signal-regulated kinase (ERK), p38MAPK and c-Jun NH2-terminal kinase (JNK), have been identified as involved in the pathogenesis of inflammatory response to extracellular signals. Compared with the control group, rapamycin treatment resulted in significantly lower activation of p38, ERK and JNK signals in the kidney of transplanted mice, indicating the inhibitory effect of rapamycin on kidney inflammation (Figure 3B).

### 3.3. Rapamycin Treatment Inhibited Lymphocyte Activation and Proliferation

Rapamycin is a carbocyclic lactone–lactam macrolide antibiotic with strong immunosuppressive properties. To investigate the cellular mechanisms by which rapamycin alleviates the disease symptoms, we explored the effect of rapamycin treatment on lymphocyte activation and proliferation in vivo. The activation/memory phenotype (CD44highCD62low) of CD4+ T lymphocytes in peripheral blood was analyzed by flow cytometry. The results shown in Figure 4A indicate that the percentage of this population in the whole blood of rapamycin-treated mice was about 11.05% ± 0.35, while that in control group mice was about 45.35% ± 7.85, indicating that the activation and migration of CD4 T lymphocytes to target organs was greatly reduced. Spleen CD4 and CD8 T cells from nephrotic mice also mainly exhibited activation/memory phenotype and expressed the early activation marker CD69. Treatment with rapamycin significantly inhibited the activation of CD4 and CD8 T cells in the spleen. This was based on the observation that the percentages of CD4+CD69+ cells and CD8+CD69+ cells in the spleen cells of rapamycin-treated mice were significantly decreased (3.6% ± 0.4 vs. 7.65% ± 0.64 and 0.97% ± 0.12 vs. 3.25% ± 0.49) compared with the control mice (Figure 4B). In addition, in rapamycin-treated mice, the expression of I-Ab in CD45R/B220+B cells was down-regulated (852.87 ± 23.82 vs. 1267.15 ± 365.50, Figure 4C), indicating that rapamycin treatment inhibited the activation of B cells.

Down-regulation of ICOS expression on splenic T lymphocytes was also observed in rapamycin-treated mice compared to control mice (Figure 4C). To further examine the effects of rapamycin treatment on lymphocyte perforation, peripheral blood lymphocyte cells were isolated from experimental mice and stained with CFSE. The labeled cells were then resuspended in culture medium with anti-CD3 mAb and anti-CD28 mAb. The proportion of CFSE+ cells proliferating at 72 h after culture was analyzed by flow cytometry. The results presented in Figure 4D show that the proliferation of CD4+, CD8+ T cells and total lymphocytes of rapamycin-treated mice was reduced by 65.66%, 63.41% and 54.52%, respectively, compared to the cells from control mice.

### 3.4. Rapamycin Treatment Inhibited Inflammatory Cytokine Production While Enhancing IL-2 Production in Periphery

Our previous experimental results showed that activated effector T cells, including Th1, Th2 and Th17 cells, were involved in the pathogenesis of this lupus nephritis model with cGVHD, at least at some point in the progression of the disease.

As mentioned above, the kidney mRNA abundance for Th1 type cytokines such as IFN-γ and IL-2 and Th2 pattern such as IL-4 and IL-10 were up-regulated in nephritic mice in contrast to normal mice. This encouraged us to assess the intracellular cytokine production in peripheral lymphocytes. The results demonstrated that, compared with those from normal mice, the effector cells from nephritic mice secreted significantly more IFN-γ, IL-4 as well as IL-10, whereas they secreted a nonsignificantly lower amount of IL-2 (data were not shown). Interestingly, similar to the results obtained from renal mRNA analyses, the frequencies of the circulating IFN-γ-producing (0.75 ± 0.21 vs. 1.8 ± 0.28, *p* < 0.05), IL-4-producing (1.2 ± 0.14 vs. 4.4 ± 0.57, *p* < 0.02) and IL-10-producing (2.3 ± 0.14 vs. 7.9 ± 0.28, *p* < 0.01) effector cells in rapamycin-treated mice were all significantly down-regulated in comparison to those in control mice (Figure 5A–C). Meanwhile, the proportion of the circulating IL-2-producing effector cells in mice with rapamycin treatment was up-regulated compared to those in control group mice (3.05 ± 0.64 vs. 1.05 ± 0.21, *p* < 0.05). In summary, rapamycin regulated peripheral cytokine production from T cells. The inflammatory cytokines were inhibited by rapamycin, while the enhancement of IL-2 production indicated that regulatory T cells (Tregs) may be involved in immunosuppression mediated by rapamycin.

### 3.5. Rapamycin Treatment Increased Treg Proportions in Periphery

Given the up-regulation of IL-2 production, we further checked the CD25 (IL-2Rα) expression on lymphocytes by flow cytometry. As shown in Figure 5E, the percentage of CD4+CD25high T cells was significantly increased in mice treated with rapamycin compared to mice from the control group (54.77% ± 2.84 vs. 26.92% ± 2.94, *p* < 0.01). Tregs are a subset of T cells characterized by CD4+CD25high cells that also express forkhead transcription factor FOXP3. They suppress conventional T cells (Tconv) and promote tolerance. There are emerging data suggesting that rapamycin induces suppression by expansion of natural regulatory T cells (nTregs), or at least by the provision of a selective survival advantage. This tempted us to evaluate the significance of Tregs in our model with rapamycin treatment. Proportions of splenic or peripheral blood CD4+FOXP3+ T cells were determined in rapamycin-treated mice as compared with control group animals. The results shown in Figure 5F demonstrate that CD4+FOXP3+ T cells in rapamycin-treated mice were significantly increased compared to the cells from the control group (6.1% ± 0.28 vs. 3.33% ± 0.78, *p* < 0.02). Taken together, these data indicate that rapamycin treatment induced Treg populations which may contribute to the amelioration of lupus nephritis.

## 4. Discussion

The recognition of the foreign major histocompatibility complex (MHC) of the F1 host by parental lymphocytes results in extensive immunopathological changes. These include two main types of GVHD: acute and chronic [24]. The typical characteristics of acute GVHD are gastrointestinal ulcer, liver dysfunction and erythematous skin injury, while the clinical manifestations of chronic GVHD are usually similar to autoimmune diseases such as systemic lupus erythematosus, Sjogren’s syndrome, scleroderma and rheumatoid arthritis [25,26]. cGVHD is the most serious, and is an increasingly common long-term complication of allogeneic stem cell transplantation (alloSCT), affecting up to 80% of patients in some series [27]. Recently, the main means of controlling acute and chronic GVHD is the preventive use of immunosuppressants to reduce or damage the function of donor T cells. However, these treatment strategies often lead to non-specific immunosuppression or other adverse side effects. Although efforts have been made to develop new anti-GVHD treatment methods, including blocking the costimulatory pathway, inducing the tolerance of donor cells, deviation of the recipient immune system towards the Th2 response, and interference with the function or expression of adhesion molecules, cytokines or T cells, cGVHD is still a serious problem. Therefore, it is necessary to better understand the induction and pathogenesis of cGVHD [27,28,29,30,31].

Injecting DBA/2 mouse lymphocytes into C57BL/6-DBA/2F1 hybrid mouse resulted in cGVHD, accompanied by persistent lymphoproliferative, hypergammaglobulinemia and systemic-lupus-erythematosus-like disease [6,9,10,11]. These human-SLE-like animals were mainly characterized by proteinuria, splenomegaly, autoantibody production and immune-complex-mediated glomerulonephritis. With respect to histopathological analysis, when proteinuria occurred in these mice, the deposition of immune complex (IC) and diffuse proliferation were observed in the mesangial regions and leukocytic infiltration in perivascular areas. At the later stage of the disease, ICs confined to the peripheral capillary loops and proliferative glomerulonephritis appeared in endothelial cells and mesangial cells, and interstitial and intraglomerular inflammation and protein deposition in dilated tubules also appeared; the severity increased with the progression of the disease. Eventually, severe interstitial inflammation, tubular necrosis, crescent formation and glomerulosclerosis occurred in the late stage of the disease. By observing the up-expression of activated markers including CD69, ICOS and I-Ab, both T and B lymphocytes were activated in nephritic mice. In addition, many cytokines, chemokines, complements and MAPK pathways involved in the pathogenesis of lupus nephritis were significantly up-regulated in nephritic mice. Most of these inflammatory mediators were present early in the kidneys of nephrotic mice with persistent albuminuria and increased as the disease progressed. In addition, our results show that serum levels of Th1-like IgG2a and Th2-like IgG1 alloantibodies increased 2 weeks after transplantation of the cell mixture prior to nephritis and persisted until onset of albuminuria in this mouse cGVHD model, suggesting an immune disorder in the early stages of SLE-like disease [2]. Consistent with this, cytokine expression of Th1 types (e.g., IFN-γ and IL-2) and Th2 types (e.g., IL-4 and IL-10) increased with disease progression.

Rapamycin, also known as sirolimus, is a macrolide antibiotic with potent immunosuppressive properties and is clinically used for the prevention of acute rejection in renal transplantations and GVHD [32,33,34,35,36]. In renal transplant patients, combinations of rapamycin with cyclosporine (CsA) lower the rate of organ failure. The frequency of biopsy-confirmed acute rejection in the following 6 months was also decreased in patients receiving rapamycin and CsA. Furthermore, rapamycin treatment also decreased frequency of moderate and/or severe histologic grades of rejection as well as the requirement for antilymphocyte antibody treatment [15]. In a recent clinical trial of rapamycin for kidney transplant patients, rapamycin proved to have an acceptable safety profile in the prevention of kidney allograft rejection [32]. In GVHD patients, therapeutic and prophylactic treatment of rapamycin has been reported [33,34]. In a randomized multicenter trial of rapamycin vs. prednisone as initial therapy for standard-risk acute GVHD, rapamycin provided similar overall initial treatment efficacy to prednisone. Patients receiving rapamycin therapy was also associated with reduced steroid exposure, greater immune suppression discontinuation and improved patient-reported quality of life [33]. A most recent study reported a phase 2 trial of GVHD prophylaxis with rapamycin, posttransplant cyclophosphamide (PTCy) and tacrolimus/mycophenolate mofetil (MMF) after peripheral blood haploidentical transplantation. In this study, rapamycin and PTCy/MMF GVHD prophylaxis not only decreased the grade II–IV acute GVHD rates after HLA-haploidentical hematopoietic cell transplantation, but also permitted hematopoietic engraftment and led to low moderate/severe chronic GVHD rates and favorable survival [34,35]. Rapamycin acts by inhibiting mammalian target of rapamycin (mTOR), a key protein kinase necessary for cell cycle progression, thereby suppressing the proliferation and clonal expansion of interleukin-2-stimulated T lymphocytes [18,19,20,21]. Unlike other commonly used immunosuppressants, rapamycin does not appear to interfere with tolerance induction [37,38,39] and permits the in vitro expansion and suppressive function of Tregs [40,41,42]. In vivo studies also demonstrated that rapamycin treatment could enhance TGFβ-dependent FOXP3 expression and generate large-scale human regulatory T cells that suppress disease in a xenogeneic model of GVHD, thus opening the door for using Tregs as a cellular therapy to prevent GVHD, graft rejection and autoimmunity [43].

In this study, we first used an oral liquid formula of rapamycin to treat SLE-like experimental lupus nephritis induced by cGVHD. Our results suggest that rapamycin treatment reduces autoantibody production, inhibits T lymphocytes and subsequent B cell activation, reduces inflammatory cytokine and chemokine production, and thereby protects renal function by manifestation with reduced proteinuria development and attenuated histological lupus nephritis. As expected, rapamycin remarkably inhibited the lymphocytes’ proliferation in the affected mice. Furthermore, consistent with the results for changes of mRNA abundance in kidney by real-time PCR analyses, intracellular cytokine staining demonstrated that rapamycin treatment remarkably diminished the secretion of cytokines from splenocytes, including Th1 type such as IFN-γ and Th2 type such as IL-4 and IL-10. However, the production of IL-2 from splenocytes in rapamycin-treated mice was much higher than that in control group animals (Figure 5A–D). Intriguingly, Tregs were also increased in mice receiving rapamycin. Previous studies have suggested that rapamycin selectively expands the peripheral Tregs, at least providing a selective survival advantage [44,45,46], and IL-2 plays a crucial role in the maintenance of natural immunologic self-tolerance through the generation and maintenance of CD4+CD25+ Tregs [47,48]. Thus, we propose that rapamycin treatment favored the Tregs population and IL-2 augmentation further expanded Tregs in vivo, thus ameliorating lupus nephritis by suppressing activated effector cells. Our study has two limitations regarding the methodology of the proteinuria test. Instead of testing 24 h urine protein, we only tested urine protein concentration at one time point per day, which may not precisely reflect the progression of kidney dysfunction. In addition, we did not examine urine creatinine concentration. The urine protein creatinine ratio has been considered an important parameter for predicting proteinuria [49,50,51], which needs to be included in our further study on lupus nephritis.

In conclusion, our studies demonstrate that, unlike other commonly used immunosuppressants, rapamycin does not appear to interfere with tolerance induction but permits the expansion and suppressive function of Tregs in vivo. Rapamycin treatment alleviated SLE-like experimental lupus nephritis by the recovery of IL-2 production, which modulated the maintenance and expansion of CD4+FOXP3+ Tregs, in turn inhibiting activated effector T cells.

## Figures and Tables

**Figure 1 biomedicines-11-00949-f001:**
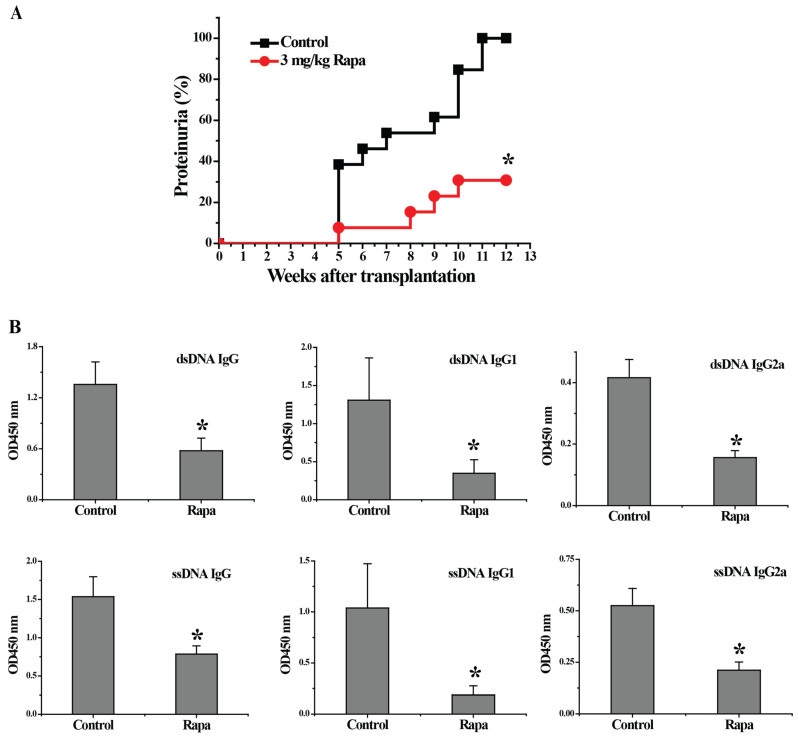
Rapamycin treatment delayed the onset of proteinuria and decreased serum autoantibodies levels in lupus-nephritis-bearing mice. (**A**) Urine was collected from week 1 to week 12 after transplantation. We found that 100% of the control group mice vs. 30.77% of the rapamycin (Rapa)-treated mice had developed proteinuria at the end of the experiments (*n* = 13 per group). (**B**) Anti-dsDNA and anti-ssDNA autoantibody levels (total IgG, IgG1 and IgG2a) in blood serum collected 3 weeks after transplantation were determined by ELISA (*n* = 6). * *p* < 0.05.

**Figure 2 biomedicines-11-00949-f002:**
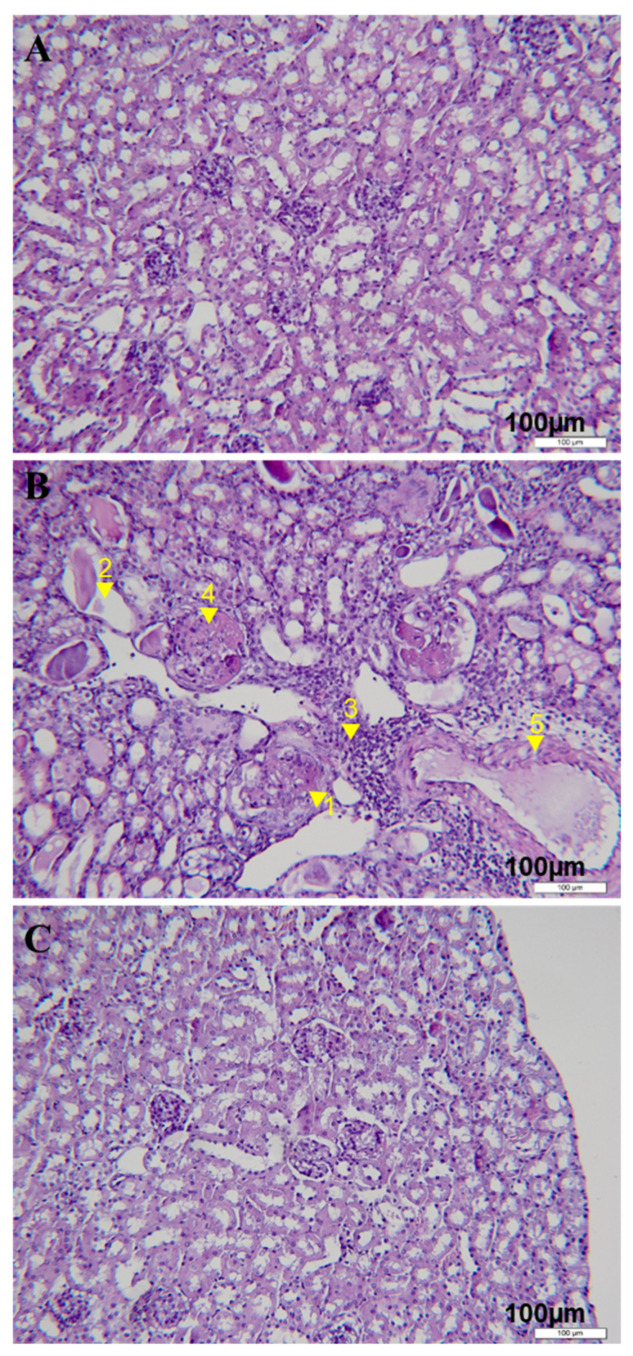
Renal histological analysis of normal mice and experimental mice treated with rapamycin or vehicle control. H&E staining on kidney tissue of normal mice without cell transfer (**A**), vehicle control-treated experimental mice (**B**) and Rapa-treated experimental mice (**C**). The 5 yellow arrows indicate mesangial proliferation (1), tubular dilation (2), leukocytic infiltration (3) and glomerular and vascular sclerosis (4, 5). Kidneys were harvested 12 weeks after transplantation, the representative images are shown.

**Figure 3 biomedicines-11-00949-f003:**
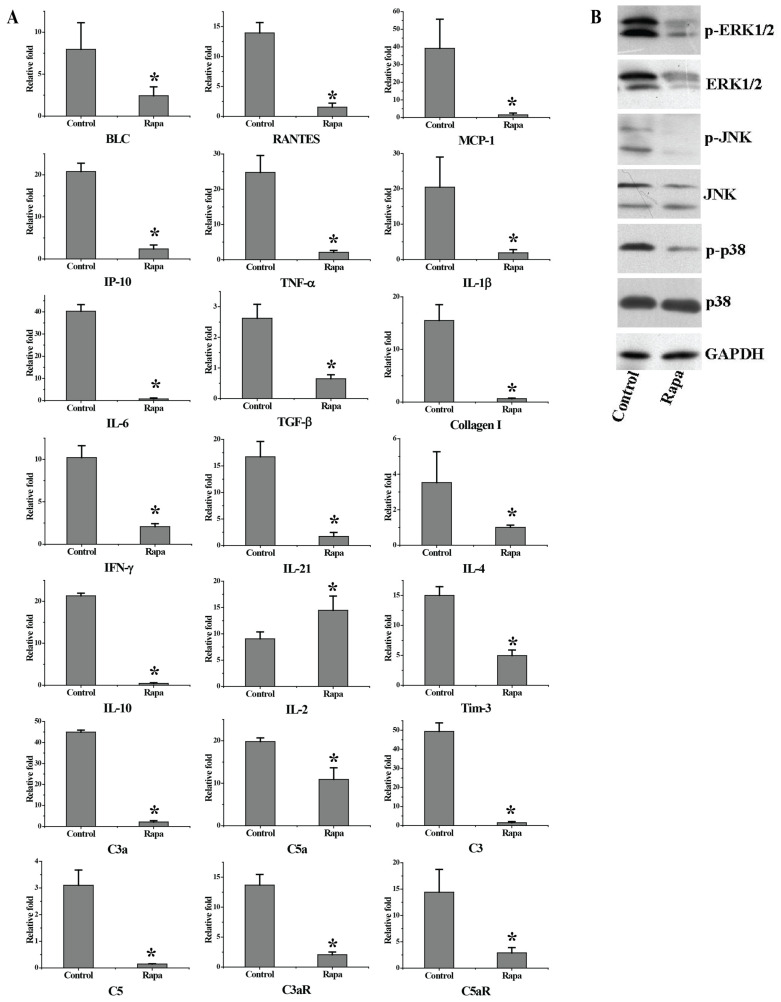
Rapamycin treatment downregulated the expression of the genes involved in pathogenesis of lupus nephritis and inflammation-related signaling pathways in the kidney. Kidneys were harvested 12 weeks after transplantation. (**A**) The renal mRNA levels of pathogenic factors related to lupus nephritis were determined by quantitative real-time PCR, (*n* ≥ 6) * *p* < 0.05. (**B**) Western blot analysis of ERK1/2 JNK p38 and GAPDH from kidney tissue. Representative date are shown from three independent experiments.

**Figure 4 biomedicines-11-00949-f004:**
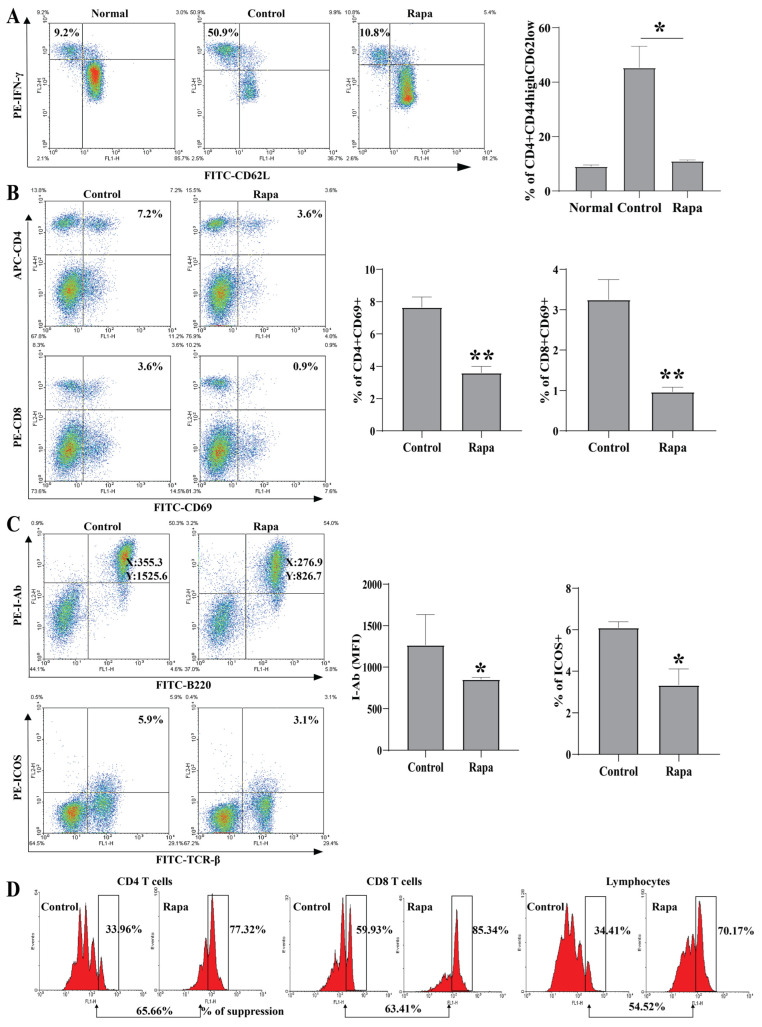
Rapamycin treatment inhibited lymphocyte activation and proliferation. (**A**) Flow analysis of CD4+CD44high CD62low population in periheral blood of the normal mice and the experimental mice (*n* = 3 per group, * *p* < 0.05). The expression of CD69 (**B**) and ICOS (**C**) on splenic T cells, and the mean fluorescence intensity (MFI) of I-Ab expression (**C**) on splenic B cells was examined by flow analysis (*n* = 3 per group, ** *p* < 0.01). (**D**) The CFSE-labeled peripheral blood lymphocytes of the experimental mice were stimulated with anti-CD3 mAb and anti-CD28 mAb for 72 h in vitro. The percentage of undivided cells in CD4 T cells, CD8 T cells and total lymphocytes was determined by flow analysis. The calculation of suppression rate was described in Section 2.9. Data are representative of at least three independent experiments.

**Figure 5 biomedicines-11-00949-f005:**
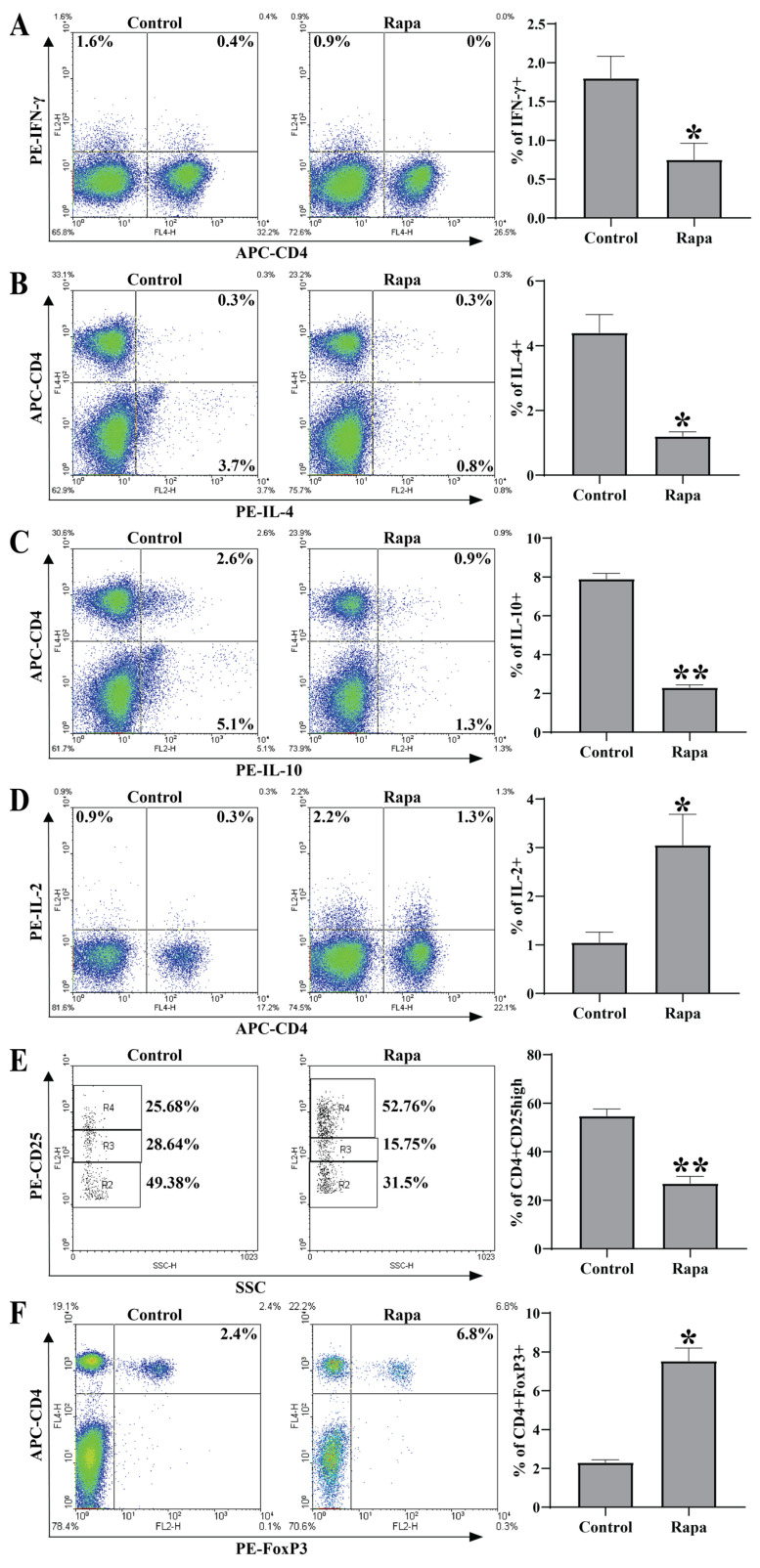
Rapamycin treatment inhibited inflammatory cytokine production while inducing IL-2 production and Tregs population in peripheral blood lymphocytes. (**A**–**D**) Intracellular cytokine staining of IFN-γ, IL-4, IL-10 and IL-2 in peripheral blood lymphocytes with PMA/ionomycin restimulation. (**E**) Flow analysis of CD25 expression on splenic CD4+ T cells. (**F**) Flow analysis of CD4+FoxP3+ Tregs in Splenic lymphocytes. *n* = 3 per group, * *p* < 0.05, ** *p* < 0.01.

**Table 1 biomedicines-11-00949-t001:** Primer sequences used for qPCR.

Genes	Primer 1 (Forward, 5′→3′)	Primer 2 (Reverse, 5′→3′)
BLC	atgaggctcagcacagcaacgctgcttc	tccgatctatgatgtttagaccgacaacag
RANTES	atgaagatctctgcagctgccctc	tgctggtgtagaaatactccttgac
MCP-1	atgcaggtccctgtcatgcttctg	ggtgatcctcttgtagctctccag
IP-10	atccctctcgcaaggacggtccgctg	tcttagattccggattcagacatc
TNF-α	attcgagtgacaagcctgtagcccac	ctgggagtagacaaggtacaacccat
IL-1β	gcccatcctctgtgactcat	aggccacaggtattttgtcg
IL-6	ttccctacttcacaagtccggagaggag	ctgcaagtgcatcatcgttgttc
TGF-β1	ataccaactattgcttcagctccacag	gtactgtgtgtccaggctccaaatat
pro (α1) I collagen	aggcaacagtcgcttcacctacagc	caatgtctagtccgaattcctggtct
IFN-γ	gcacagtcattgaaagcctagaaagtc	ggtagaaagagataatctggctctg
IL-21	agacattcatcattgacctcgtg	gaatcatcttttgaaggagcca
IL-4	ctccatgcttgaagaagaactctag	atttgcatgatgctctttaggct
IL-10	actgctaaccgactccttaatg	catcctgagggtcttcagcttct
IL-2 [21]	cccacttcaagctccacttc	atcctggggagtttcaggtt
Tim-3	cacttgtgcctatgtgctg	ttgttgagatcgcccttta
C3a	tcggaagtgttgtgaggat	gttgcagcagtctatgaagg
C5a	agaaatgctgctatgacgga	ttcttttcggatcttgttcg
C3	gggagaagttcggcatag	gttgttgaaggcagcatag
C5	gctgcctgactcactaac	tcctcgcacaacagaata
C3aR [21]	taaccagatgagcaccacca	tgtgaatgttgtgtgcatgg
C5aR [22]	gatgccaccgcctgtatagt	acgaaggatggaatggtgag
GAPDH	tgaaggtcggtgtgaacggatttg	gttgaatttgccgtgagtggagtc

## Data Availability

The data presented in this study are available on request from the corresponding author.

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
