# Peer review of "Rapamycin Treatment Alleviates Chronic GVHD-Induced Lupus Nephritis in Mice by Recovering IL-2 Production and Regulatory T Cells While Inhibiting Effector T Cells Activation"

_biomedicines, 2023, doi:10.3390/biomedicines11030949_

Round 1
Reviewer 1 Report
The authors performed a series of very interesting experiments demonstrating the immunosuppressive activity of rapamycin, a mTOR inhibitor molecule traditionally used in solid organ transplantation, in chronic GVHD induced lupus nephritis.
The paper is clear, experiments well described and illustrate, the references updated.
There are just few points to clarify:
1. In the section 3.3 evaluating peripheral blood lymphocytes, the authors reported a reduction of CD4 activated/memory T cells in rapamycin treated mice compared to control mice group. There was also a reduction of CD8 cells?
2. What were the lymphocyte number in the control group? Did they observed reduced CD4 numbers compared to mice without GVHD?
3. Have the authors idea of the numbers of B regulatory cells?
4. They reported a reduction of auto-antibodies production in rapamycin treated mice. How about total antibody level? There was an impact on infections control?
5. Are there any differences compared to sirolimus activity? Please add a comment in discussion.
Author Response
- In the section 3.3 evaluating peripheral blood lymphocytes, the authors reported a reduction of CD4 activated/memory T cells in rapamycin treated mice compared to control mice group. There was also a reduction of CD8 cells?
Response: Yes, in figure 1B, we show both activated (CD69+) CD4 and CD8 T was decreased.
- What were the lymphocyte number in the control group? Did they observed reduced CD4 numbers compared to mice without GVHD?
Response: Compared to normal mice without GVHD, the mice without rapamycin treatment (control group) have higher frequency of effector memory CD4 T cells population (Figure 4A), while rapamycin treatment inhibits this population.
- Have the authors idea of the numbers of B regulatory cells?
Response: We thank the review’s suggestion on B regulatory cells, in the present study, we only test impact of rapamycin the total B cells activation (figure 4C), we will test the effect of rapamycin on B regulatory cells in future study.
- They reported a reduction of auto-antibodies production in rapamycin treated mice. How about total antibody level? There was an impact on infections control?
Response: Our data (figure 4C) and other’s study has shown that Rapamycin inhibit B cell proliferation and activation and blocked B cell differentiation into plasma cells (PMID: 26087255). So, we suspect that total antibody level would be decreased. We only tested the total serum dsDNA and ssDNA IgG which are the biomarkers of lupus nephritis progression.
Rapamycin is an antifungal antibiotic and immunosuppressant; we expect that mice received rapamycin may have better fungal infections control, but not the other infections. Indeed, we did not observe any symptoms of other infection during experiments. Since our study focused on lupus nephritis, we did not test whether the mice are resistant to infections or not.
- Are there any differences compared to sirolimus activity? Please add a comment in discussion.
Response: The rapamycin used in this study is an oral liquid formulation purchased from NCPC New Drug Research and Development Co.Ltd. Since sirolimus is rapamycin, we believed there would be no difference between rapamycin we used and sirolimus activity. We mention sirolimus in line 319.
Reviewer 2 Report
The article deals with effects of a well-known immune suppressor rapamycin in experimental model of lupus-like nephritis associated with chronic GVHD (as defined by authors) induced in mice by immunization with MHS-incompatible lymphocyte mixtures. Numerous, quite appropriate biomarkers of cellular and humoral immunity, and autoimmune markers were assayed in treated controls and with rapamycin. In particular, the authors showed reduced autoantibody levels, less pronounced pathological changes in affected kidneys, increase of Treg cell populations in experimental animals treated with rapamycin as well as downregulation of several inflammation-controlling genes. Hence, the rapamycin effects were correctly evaluated, and these results confirm immunosuppressive action of rapamycin in this autoimmune condition.
Remarks:
Lane 65-66. … Donor cells were injected intravenously (i.v.) four times at 3- to 4-day intervals within two weeks. One should describe the schedule of Rapamycin therapy (at what time point was it started in relation to immunization schedule).
In the Introduction (since lane 40) and in Discussion (lanes 291 and further on): the tested SLE-like murine model using inter-strain immunization with mixed lymphocytes was developed decades ago (in 80’s) and today it does not correspond to current definition of the term “chronic GVHD” observed after hematopoietic stem cell transplantation. Moreover, such kidney damage is not common in either acute or chronic GVHD. Therefore, the author should modify the conclusions (including Abstract and Introduction) concerning cGVHD alleviation, and limit their conclusions to modulatory effects of rapamycin in SLE-like experimental nephritis.
The paper deserves publication, however, after clarifying terminology and technical details of experimental model. Minimal copy editing could be done.
Author Response
Lane 65-66. … Donor cells were injected intravenously (i.v.) four times at 3- to 4-day intervals within two weeks. One should describe the schedule of Rapamycin therapy (at what time point was it started in relation to immunization schedule).
Response: The schedule of Rapamycin treatment was described from line 97-100.
“Mice received DBA/2J donor lymphocytes were randomized to receive either Rapamycin (3 mg/kg, n=13) or equal volume of saline as control treatment group (n=13) once daily by oral gavage post transplantation, which were successive till the end of the experiments.”
In the Introduction (since lane 40) and in Discussion (lanes 291 and further on): the tested SLE-like murine model using inter-strain immunization with mixed lymphocytes was developed decades ago (in 80’s) and today it does not correspond to current definition of the term “chronic GVHD” observed after hematopoietic stem cell transplantation. Moreover, such kidney damage is not common in either acute or chronic GVHD. Therefore, the author should modify the conclusions (including Abstract and Introduction) concerning cGVHD alleviation, and limit their conclusions to modulatory effects of rapamycin in SLE-like experimental nephritis.
Response: We thank the review’s comments and modify the abstract and conclusion related to cGVHD alleviation and limit the conclusions to modulatory effects of rapamycin in SLE-like experimental lupus nephritis.
Reviewer 3 Report
The manuscript "Rapamycin treatment alleviates chronic GVHD-induced lupus nephritis in mice by recovering IL-2 production and regulatory T cells while inhibiting effector T cells activation", written by Zhang J, Wang X, Wang R, Chen G, Wang J, Feng J, Li Y, Yu Z and Xiao H. presents the effects of rapamycin treatment on the autoimmune disease development and its consequences, on the model of mice receiving allogenic lymphocytes. Rapamycin delayed the onset of proteinuria and levels of serum autoantibodies, as well as renal tissue damage. The results also showed downregulation of genes involved in inflammation and changes in cytokine expression, as well as changes in lymphocyte activation and proliferation.
The experiments are in general well done and support the hypothesis of rapamycin attenuation of chronic graft versus host disease. However, several issues need additional explanations and corrections. In Material and methods, cell costimulation should be better explained. In Results, short explanation of the whole experiment is needed, not just the final results. Figure legends should not give the discussion of the experiments, the values of standard deviation could be omitted and significance of the results signed by asterisk. On the other side, i.e. Fig. 4 needs additional explanations of the flow cytometry methods used, especially Fig.4. D. In the Discussion, there is a repetition of the results, even with detailed values. On the other side, discussion on clinical trials and data from patient therapy are just mentioned. These data should be better explained. The most recent reference is from 2011.
Other comments
sentence reorganization: lines 82, 119,137, 174, 184, 187,193, 207, 219, 221, 240, 251
line 102: Table 1
line 110: 1 mM
line 114: antibodies
line 164: explanation of control treatment
Author Response
The experiments are in general well done and support the hypothesis of rapamycin attenuation of chronic graft versus host disease. However, several issues need additional explanations and corrections. In Material and methods, cell costimulation should be better explained.
Response: We are sorry for the confusing sentence. Anti-CD28 mAb is commonly used to stimulate CD28 co-simulation signaling in T cells. We now simplify the sentence to avoid confusion. Lin201-line205
In Results, short explanation of the whole experiment is needed, not just the final results.
Response: We thank review for this comment, we now add explanation of the whole experiment before introducing the results. Lin 225-230
Figure legends should not give the discussion of the experiments, the values of standard deviation could be omitted and significance of the results signed by asterisk.
Response: We thank for review’s suggestion; we have updated all the figure legend accordingly.
On the other side, i.e. Fig. 4 needs additional explanations of the flow cytometry methods used, especially Fig.4. D.
Response: We are sorry for the unclear figure legend; We have updated the figure legend for Fig.4.
In the Discussion, there is a repetition of the results, even with detailed values. On the other side, discussion on clinical trials and data from patient therapy are just mentioned. These data should be better explained. The most recent reference is from 2011.
Response: We thank for review’s comments, the repetition of the results in Discussion section is removed. The clinical studies with rapamycin treatment were discussed and the recent studies was cited. (Lin497-510). The added literatures are highlighted in yellow in References section.
Other comments
sentence reorganization: lines 82, 119,137, 174, 184, 187,193, 207, 219, 221, 240, 251
Response: These sentences have been reorganized.
line 102: Table 1
Response: Fixed
line 110: 1 mM
Response: Fixed
line 114: antibodies
Response: Fixed
line 164: explanation of control treatment
Response: The control treatment has been explained at the beginning of this paragraph. Lin229.
Reviewer 4 Report
The manuscript submitted by Zhang et al, entitled "Rapamycin treatment alleviates chronic GVHD-induced lupus nephritis in mice by recovering IL-2 production and regulatory T cells while inhibiting effector T cells activatio" aims to show that unlike other commonly used immuno-suppressants, the agent Rapamycin does not appear to interfere with tolerance induction but allows the expansion and suppressive function of Treg in vivo. Rapamycin treatment imrpoves the cGVHD-induced Lupus nephritis in mice by recovery of IL-2 production that modulated the maintenance and expansion of CD4+FOXP3+ regulatory T cells which inhibited activated effector T cells.
The manuscript is well-written, with the M&M well-described and results in some points nicely discussed. However, the following aspects should be improved before its resubmition:
1- Line 81: please justify the used threshold;
2- In the M&M section, all reagents and kits should have the reference descriminated;
3- Line 159: please justify why only 30.77% of the mice treated with Rapamycin, showed improved proteinuria;
4-Figure 2: why authors only conducted the study on the kidneys until 12 weeks. A more longer study will be amazing;
5-line 187, please discuss the limitations of qRT-PCR
6- There is any information about the cytotoxicity of Rapamycin, at the used dosis, in mouse hepatocytes?
7- Figure 3, B - An entire image of whole blots should be presented as supplementary data.
Author Response
1- Line 81: please justify the used threshold;
Response: The urine protein concentration of 100 mg/dl is determined based on published study (PMID: 33883349, this literature is now cited in the text). This study reports that urinary protein concentration is 73.7 ± 8.2 mg/dl in wild-type naïve mice. We believe 100 mg/dl should be sufficient to determine whether the mice develop proteinuria.
2- In the M&M section, all reagents and kits should have the reference descriminated;
Response: This information was added for all reagents and kits.
3- Line 159: please justify why only 30.77% of the mice treated with Rapamycin, showed improved proteinuria.
Response: We are sorry for confusion description. We have updated the sentences and make it clear.
“All the control mice had proteinuria at 12 weeks after cell transfer, while only 30.77% of rapamycin-treated mice developed proteinuria. This data suggested that rapamycin treatment significantly delayed the onset of proteinuria.”
4-Figure 2: why authors only conducted the study on the kidneys until 12 weeks. A more longer study will be amazing;
Response: We thank for review’s suggestion. The incidence rate of proteinuria is ready 100% in control group 12 weeks after cell transfer, but we agree that it is worth to test the effect of rapamycin in a longer period. We will test this hypothesis in our future study.
5-line 187, please discuss the limitations of qRT-PCR
Response: The discussion on the limitations of qRT-PCR is added (line 290-294)
6- There is any information about the cytotoxicity of Rapamycin, at the used dosis, in mouse hepatocytes?
Response: The commonly used dose of rapamycin in mouse study is 3~8mg/Kg/Day (PMID: 20146790, PMID: 24312548; PMID: 35958488; PMID: 27549339). A study also shows that rapamycin injection at 8 mg/kg/day for 3 months extends life expectancy of male mice (PMID: 27549339). We choose the lowest dose (3mg/kg/day) to avoid potential cytotoxicity, and we did see any liver tissue damage when perform autopsy.
7- Figure 3, B - An entire image of whole blots should be presented as supplementary data.
Response: The unmodified whole blots was added as supplementary data.
Reviewer 5 Report
Jilu Zhang et al. have carried out a complex preclinical study in which they have verified that the administration of the antibiotic rapamycin in mice ameliorates chronic graft-versus-host disease (cGVHD)-induced lupus nephritis. It is a very complete work that includes a great variety of analytical techniques and evaluated parameters. Also, overall, it is well written, designed, and presented.
However, there are some aspects that should be reviewed and resolved before the final publication of the manuscript:
- Section 2.1. I think it is essential to explain in detail the methodology for collecting the serum and urine samples (mainly the latter: was an isolated urine sample collected? Was the urine excreted for 24 hours collected?
- Section 2.3. Why was work done with the concentration of protein in urine? On many occasions, the urinary concentration of a solute is not at all representative of the physiological or pathological state in which an individual is, since the urine may include a greater or lesser amount of water, affecting the concentration. I think that the correct thing is to work with the amount of solute excreted per unit of time (and more specifically, I think that the most correct thing would be to express the biomarker in the amount excreted/24 hours).
- Lines 59-61: I think that no description of the results of the study should appear in the introduction, since the introduction should present the background of the topic.
- In general, throughout the text (but especially in section 2.7.) there are numerous abbreviations that are not described.
- Section 2.10: Why was Student's t-test directly selected? I think that first it is necessary to demonstrate that the data to be analyzed in both study groups are normal. Otherwise, another statistical test would have to be selected.
- Figures: I understand that the statistical significance is presented in the figure caption, but it is very useful to insert symbols in the graphs to indicate the statistical differences between the groups and their degree of significance.
- Regarding Figure 2, I don't understand why the photos of the Control group are larger. Nor is the scale of the photographs specified and whether they all have the same magnification (it seems not). The histological alterations detected are also not indicated with arrows of different colors in the photographs.
- The Bibliography section does not include any recent references (for example, from the last 10 years). I think it would be appropriate if most of the bibliographical sources used were relatively recent.
- There are some words that repeatedly appear written with the first letter capitalized (such as rapamycin, lupus, flow and intracellular). I think they should be written in lowercase, since they are not proper names or trademarks.
- Line 53: Streptomyces hygroscopicus should be written in italics.
- Line 67: What does the "F." before "Serum" mean?
- Line 74: I think it should be "Mice that received..."
- Line 102: I think it should be "Table"
- Line 126: The official abbreviation for hours is h.
- Line 152: What does the symbol after "P" mean?
Author Response
- Section 2.1. I think it is essential to explain in detail the methodology for collecting the serum and urine samples (mainly the latter: was an isolated urine sample collected? Was the urine excreted for 24 hours collected?
Response: We are sorry for insufficient description of the methodology for serum and urine collection. We have updated the Material and Method section for serum and urine collection.
The mouse blood was collected by cutting the tail. The serum was collected by centrifuging the blood at 1,000-2,000 x g for 10 minutes in a refrigerated centrifuge.
For urine collection, it is not 24-hour collection. we use the method from published study (studies (PMID: 33883349, PMID: 28337988). Briefly, urine was collected from spontaneous urination when handled the mice. In case that mice don’t urinate, we will give a gentle pressure on the bladder. The urine was collected in 1.5mL Eppendorf tubes and kept on ice until testing. The protein concentration was determined within 2 hours after collection.
- Section 2.3. Why was work done with the concentration of protein in urine? On many occasions, the urinary concentration of a solute is not at all representative of the physiological or pathological state in which an individual is, since the urine may include a greater or lesser amount of water, affecting the concentration. I think that the correct thing is to work with the amount of solute excreted per unit of time (and more specifically, I think that the most correct thing would be to express the biomarker in the amount excreted/24 hours).
Response: We thank for the reviewer’s suggestions and totally agree with the reviewer that testing the amount of urine solute excreted in 24hr is the more accurately way to determine the proteinuria. As our animal facility is not equipped with metabolic cages, we are unable to perform 24-hour urine collection.
The urine analysis we performed in our study has been used in published mouse studies (PMID: 33883349, PMID: 28337988). Furthermore, in order to avoid false positive results, when we detect high urine protein concentration, we will collect urine twice again in the following 24 hour and confirm the proteinuria. This detail was added in Material and Method section 2.3.
- Lines 59-61: I think that no description of the results of the study should appear in the introduction, since the introduction should present the background of the topic.
Response: We agree with the review and modified the introductions and remove the description of the results.
- In general, throughout the text (but especially in section 2.7.) there are numerous abbreviations that are not described.
Response: We have spelled out the abbreviations.
- Section 2.10: Why was Student's t-test directly selected? I think that first it is necessary to demonstrate that the data to be analyzed in both study groups are normal. Otherwise, another statistical test would have to be selected.
Response: We thank the review for the suggestion on statistical analysis. In our study, the age and gender matched mice randomly distributed into rapamycin treated group and control group. When we perform the assay on mice from both groups, the timing and procedures are identical, so, the generated data from the mice should conform to normal distribution. Therefore, we choose Unpaired t- test, which is also the most used statistical analysis in published mouse studies that comparing the difference between two groups.
- Figures: I understand that the statistical significance is presented in the figure caption, but it is very useful to insert symbols in the graphs to indicate the statistical differences between the groups and their degree of significance.
Response: We thank the review for the suggestion and make the changes in all figures.
- Regarding Figure 2, I don't understand why the photos of the Control group are larger. Nor is the scale of the photographs specified and whether they all have the same magnification (it seems not). The histological alterations detected are also not indicated with arrows of different colors in the photographs.
Response: We are sorry for missing the scale bars and indication of histological alterations. To avoid confusion, we now use the original pathology images (The previous fig.2 images were cropped from the original pathology images) with scale bars for figure 2. We also add 5 arrows that indicates the pathology changed.
Mice from the controls group develop severe Lupus nephritis, characterized by mesangial proliferation, tubular dilation, leukocytic infiltration in perivascular areas and in interstitial areas, tubular atrophy, as well as glomerular and vascular sclerosis. This pathology changed make the glomerular look larger and perivascular areas look wider when compared to normal mice and rapamycin-treated mice.
- The Bibliography section does not include any recent references (for example, from the last 10 years). I think it would be appropriate if most of the bibliographical sources used were relatively recent.
Response: We thank the review for the suggestion and added the recent literatures of rapamycin in GVHD and renal transplant. The added literatures are highlighted in yellow in References section.
- There are some words that repeatedly appear written with the first letter capitalized (such as rapamycin, lupus, flow and intracellular). I think they should be written in lowercase, since they are not proper names or trademarks.
Response: These words with first letter capitalized have been changed accordingly.
- Line 53: Streptomyces hygroscopicus should be written in italics.
Response: Fixed
- Line 67: What does the "F." before "Serum" mean?
Response: We are sorry for the confusion, the “F” is removed.
- Line 74: I think it should be "Mice that received..."
Response: Fixed
- Line 102: I think it should be "Table"
Response: Fixed
- Line 126: The official abbreviation for hours is h.
Response: Fixed
- Line 152: What does the symbol after "P" mean?
Response: We are sorry for the confusion, it should be “p<0.05”. we have changed to “p”.
Round 2
Reviewer 3 Report
The authors of the manuscript "Rapamycin treatment alleviates chronic GVHD-induced lupus 2 nephritis in mice by recovering IL-2 production and regulatory 3 T cells while inhibiting effector T cells activation" responded to the comments made on the first version of the text. I do not have further comments, except several minor observations:
in the Abstract, full name of abbreviation SLE
line 105: was
line 106: units should be separated from numbers
line 157: proteins were prepared
line 164: the... was sealed in the milk?
line 200: FACSCalibur
line 215: hours
line 288: downregulated
line 494: rate?
line 495: following?
line 550: receiving
Author Response
Thanks for the reviewer’s comments, we have fixed these minor issues.
in the Abstract, full name of abbreviation SLE
Fixed
line 105: was
Fixed
line 106: units should be separated from numbers
Fixed
line 157: proteins were prepared
Fixed
line 164: the... was sealed in the milk?
The sentence has been reorganized.
line 200: FACSCalibur
Fixed
line 215: hours
Fixed
line 288: downregulated
Fixed
line 494: rate?
The sentence has been reorganized.
line 495: following?
The sentence has been reorganized.
line 550: receiving
Fixed
Reviewer 4 Report
The authors add the details asked by this reviewer, as requested. Thus, the manuscript reached a suitable form to be publish.
Author Response
We thank for the reviewer’s suggestion.
Reviewer 5 Report
The article has improved remarkably and, in general, my suggestions and doubts have been resolved by the authors. However, I think there are two fundamental aspects that should still be reviewed:
- According to the authors, they have not been able to collect 24-hour urine due to the lack of metabolic cages. In that case, I think the right thing to do is to quantify the creatinine concentration in those urine samples and present the proteinuria results as amount of protein/amount of creatinine. In this way, it is possible to "correct" slightly the possible differences in volume and concentration that the samples may present (as is done in some studies with patients).
- The authors indicate that they selected the Student's t statistical test because, according to the methodology used, the data should conform to normal distribution and because it is the most common test in studies with mice. This is not an acceptable justification for selecting such a test. The normality of the data from both study groups must be previously evaluated with an appropriate test and, depending on the results, the Student's t test or the Mann-Whitney test must be selected.
Author Response
- According to the authors, they have not been able to collect 24-hour urine due to the lack of metabolic cages. In that case, I think the right thing to do is to quantify the creatinine concentration in those urine samples and present the proteinuria results as amount of protein/amount of creatinine. In this way, it is possible to "correct" slightly the possible differences in volume and concentration that the samples may present (as is done in some studies with patients).
We thank the review for the constructive suggestion on urine creatinine test, and completely agree that the urine protein creatinine ratio has been considered as an important parameter for predicting kidney dysfunction. During the 12-week experiment, our urine test was performed with in 2 hours after urine collection, we now do not have any fresh or frozen urine samples for testing creatinine.
Given that we are unable to test creatinine in our urine samples, we have described these study limitations (including the method of urine collection and creatinine test ) in the in discussion section (Line 565-671).
- The authors indicate that they selected the Student's t statistical test because, according to the methodology used, the data should conform to normal distribution and because it is the most common test in studies with mice. This is not an acceptable justification for selecting such a test. The normality of the data from both study groups must be previously evaluated with an appropriate test and, depending on the results, the Student's t test or the Mann-Whitney test must be selected.
Thank you for your constructive comment and we are sorry that the overly succinct description on Section: “statistical analysis” has caused misunderstanding. Actually, for each dataset, the normality of data was firstly check using Shapiro-Wilk test followed by test on homogeneity variance. Only those datasets meet the standard of normal distribution and homogeneity was further analyzed using Student’s t-test, otherwise, Mann Whitney U test was selected. We have re-write the section of Statistical Analysis. Line 230-236